

# Sex differences in peripheral monoamine transmitter and related hormone levels in chronic stress mice with a depression-like phenotype

Yitian Chen[1,2,*], Weijia Cai[1,2,*], Canye Li[1,2], Zuanjun Su[1,2], Zhijun Guo[2], Zhuman Li[2], Chen Wang[2] and Feng Xu[1,2]

[1] School of Pharmaceutical Sciences, Southern Medical University, Guangzhou, China
[2] Department of Clinical Pharmacology, Sixth People's Hospital South Campus, Shanghai Jiaotong University, Shanghai, China
[*] These authors contributed equally to this work.

Corresponding author
Feng Xu, andrewfxu1998@gmail.com, xuf@smu.edu.cn

## ABSTRACT

**Backgrounds**. Chronic stress could induce depression-like phenotype in animal models. Previous data showed that sex differences exist after chronic stress model establishment, however, the detailed information about the difference of blood biochemical indexes is not clear. In this study, we aim to supply comparison of monoamine transmitters and related hormone markers in serum between male and female depressed mice, and in order to better understand the sex difference in transmitters and hormone levels in depression occurrence and development.

**Methods**. Sixty C57BL/6 mice (both male and female) were divided into two groups by gender. Same gender mice were then divided randomly into the non-treated control group and chronic stress group which was exposed to 8 weeks of chronic unpredictable mild stress (CUMS). Depression-like behavior was assessed with open-field test and sucrose preference test. Blood sample was collected and monoamine transmitter and related hormone in serum were measured by ELISA.

**Results**. The depression-like phenotype mice model was established successfully after 8 weeks of chronic stress. The locomotion activity scores in male stressed mice declined more than that in female stressed mice, while the exploratory behavior scores in female stressed mice declined more than that in male stressed mice. Compared to non-treated control group mice, mice in the chronic stress group in response to stress showed greater declines in monoamine transmitters (5-HT, dopamine, norepinephrine) and sex hormones (androgen, estrogen, oxytocin and prolactin), while stress hormones (adrenaline, corticosterone and ACTH) were significantly increased. The decrease of norepinephrine, androgen and estrogen in female stressed mice was greater than in male stressed mice, whereas the 5-HT and oxytocin in male stressed mice decreased more than in female stressed mice, and the corticosterone in male stressed mice increased more than in female stressed mice.

**Conclusion**. Sex differences of monoamine transmitter and related hormone levels in serum occurred in chronic stress induced depression-like phenotype mice model. It may provide a useful reference to guide precise antidepressant treatment in different gender population in clinical care.

## INTRODUCTION

Depression is a global health priority. As the third cause of disease burden worldwide at present, depression will be ranked first by 2030 (*Malhi & Mann, 2018*). According to WHO, depression accounts for 10% of the total nonfatal disease burden. Unfortunately, this burden falls disproportionately on girls and women (*Parker & Brotchie, 2010*). The prevalence of major depression in women is as twice as in men (*Salk, Hyde & Abramson, 2017*; *Sassarini, 2016*). Epidemiological research demonstrated a substantial sex-related differences in prevalence for specific mental disorders (*Liu et al., 2019*). Even in the same age group the incidence of depression in females was higher than that in males (*Barnett et al., 2021*; *Euteneuer et al., 2011*; *Seifert et al., 2021*). Females seemed to be more susceptible to stress via the hypothalamus pituitary adrenal (HPA) axis and leading to abnormal hormone secretion (*Blier & El Mansari, 2013*; *Kolbasi et al., 2021*; *Mainio et al., 2006*; *Zefferino, Di Gioia & Conese, 2021*). In addition, from the perspective of social psychology, females were more challenged by job and family stress which might aggravate depressive disorder (*Isaksson et al., 2020*; *Maji, 2018*; *Wang et al., 2020*). Generally, women showed higher risk of mood and anxiety disorders (internalizing disorders), meanwhile men showed higher rates of antisocial and substance abuse disorders (externalizing disorders) (*Cavanagh et al., 2017*; *Eaton et al., 2012*).

In our previous work, we established chronic stress induced depression-like phenotype both in female and male mice/rat models (*Xia et al., 2016*; *Yang et al., 2021*; *Zeng et al., 2014*; *Zhou et al., 2021*). As in other research reports, we have noticed sex differences in depression-like behavior (*Dalla et al., 2011*; *Goodwill et al., 2019*; *Huynh et al., 2011*; *Leussis et al., 2012*; *Liu et al., 2019*). We speculate that the sex differences in depression-like behavior might be related to the difference secretion of neurotransmitter, stress hormones and sex hormones. Despite some sporadic reports on serum biochemical molecules (*Bao et al., 2004*), few comprehensive studies have been carried out to compare the difference of transmitter and hormone between female and male. Based on comparing the differences in secretion of several peripheral monoamine transmitters and related hormone markers in chronic stress male and female mice, this study aimed to provide useful information for the diagnosis and treatment of depression in men and women in clinical practice.

## MATERIALS AND METHODS

### Mice

Sixty C57BL/6 mice (male/female) of 3–4 weeks weighing 11 g–18 g which were healthy SPF mice without gene modification were purchased from Shanghai Jiesijie Experimental Animal Co., Ltd. (License no.: SCXK (Shanghai) 2013-0006), and housed in a temperature-controlled room (22 ± 2 °C) and humidity-controlled (55 ± 5%) on a 12 h/12 h light/dark cycle with access to standard chow diet and water ad libitum. After 1 week of acclimatization,

mice with same gender were randomly divided into two groups ($n = 15$): non-treated control group (fed normally without any stress) and chronic stress group (exposed to 8 weeks of chronic unpredictable mild stress). We selected random number generator to generate random seed number for grouping. Animal procedures were complied with the ARRIVE guidelines and performed in accordance with the guidelines of the U.K. The Animal (Scientific Procedures) Act 1986 and associated guidelines, EU Directive 2010/63/EU for animal experiments. The animal experiment protocol was approved by the Bioethics Committee of Fengxian Hospital, Southern Medical University (approval number: 22010737).

## Chronic stress model

Chronic unpredictable mild stress (CUMS) was a classical depression-like behavior/phenotype animal model. Mice were exposed to a series of stressors over several week to induce a range of behavioral alterations including apathy and anhedonia. Those behavioral changes fairly simulate the behavioral characteristics of patients with clinical depression (*Katz, 1982*; *Willner et al., 1987*). Male/female mice in the chronic stress group were housed individually and exposed to 8 weeks of chronic unpredictable mild stress (CUMS) once a day as previously described (*Zeng et al., 2014*). The same stressor was not allowed to use consecutively. Mice were moved to an isolated room when exposed to stress, and moved back after stress treatment. Meanwhile, male/female mice in the control group were raised in the same experimental environment without any stress. The subjected chronic stressor patterns are shown in Table 1.

## Behavior assessment

Depression-like phenotypes were assessed with the open field test (OFT) (*Walsh & Cummins, 1976*) and sucrose preference test (SPT) (*Willner et al., 1987*). The OFT of each mouse was conducted in a quiet room with Animal Behavior Trajectory Video Analysis System (Labmaze, ZSDC Science Technology Co., Ltd., China) after CUMS model establishment. Mouse was placed individually in the center of the animal behavior track analysis instrument (100 cm × 100 cm × 40 cm) which were divided into 25 grids ($20 \times 20$ cm$^2$). The number of squares crossed by each mouse within 5 min was recorded as its locomotion activity scores, and the number of standing times of each mouse was recorded as its exploratory behavior scores.

The sucrose preference test (SPT) procedure was also performed after CUMS model establishment. Briefly, mice were individually exposed to 1% sucrose solution (w/v) for the first 24 h. Then one bottle of 1% sucrose solution and another bottle of tap water were presented at random positions to mice for the next 24 h, followed by food and water deprivation for the third 24 h. On day 4, two bottles were weighed and presented to each mouse for 4 h. The position of the two bottles was set randomly. The sucrose preference was calculated by the equation as follow: Sucrose preference = sucrose consumption (g)/(water consumption (g) + sucrose consumption (g)) ×100%.

**Table 1   Chronic stress model establishment procedure.**

| Week | Sunday | Monday | Tuesday | Wednesday | Thursday | Friday | Saturday |
|---|---|---|---|---|---|---|---|
| 1 | Noise interference 1 h | Day/night inversion 24 h | Cage tilting (45°) 24 h | Wet bedding 24 h | Horizontal shaking 10 min | Restraint stress 1 h | Day/night inversion 24 h |
| 2 | Clip tail 1 min | Restraint stress 1 h | Cage tilting (45° ) 24 h | Wet bedding 24 h | Hot water swimming (45 °C) 5 min | Coldwater swimming (4 °C) 5 min | Cage tilting (45° ) 24 h |
| 3 | Horizontal shaking 10 min | Cage tilting (45°) 24 h | Horizontal shaking 10 min | Day/night inversion 24 h | Cage tilting (45°) 24 h | Hot water swimming (45 °C) 5 min | Clip tail 1 min |
| 4 | Wet bedding 24 h | Restraint stress 1 h | Day/night inversion 24 h | Noise interference 1 h | Cage tilting (45°) 24 h | Noise interference 1 h | Restraint stress 1 h |
| 5 | Hot water swimming (45 °C) 5 min | Cold water swimming (4 °C) 5 min | Day/night inversion 24 h | Noise interference 1 h | Wet bedding 24 h | Noise interference 1 h | Coldwater swimming (4 °C) 5 min |
| 6 | Clip tail 1 min | Horizontal shaking 10 min | Restraint stress 1 h | Clip tail 1 min | Cold water swimming (4 °C) 5 min | Clip tail 1 min | Horizontal shaking 10 min |
| 7 | Restraint stress 1 h | Clip tail 1 min | Wet bedding 24 h | Hot water swimming (45 °C) 5 min | Noise interference 1 h | Wet bedding 24 h | Cage tilting (45° ) 24 h |
| 8 | Clip tail 1 min | Day/night inversion 24 h | Noise interference 1 h | Wet bedding 24 h | Cage tilting (45°) 24 h | Clip tail 1 min | Wet bedding 24 h |

## Peripheral blood sample collection and determination

Before and after CUMS model establishment and behavioral assessment, mice were kept unconscious by intraperitoneal injection of 1% sodium pentobarbital 0.1 ml/20 g. The orbital venous plexus blood of mice was collected by glass capillary at 8:00 am. All blood samples were placed in a coagulation tube and allowed to clot for 2 h at room temperature before $1000 \times$ g for 15 min. Then the upper serum was collected for assaying the level of 5-HT, dopamine, norepinephrine, androgen, estrogen, oxytocin and prolactin, adrenaline, corticosterone and ACTH. The ELISA kits of 5-HT, dopamine, norepinephrine, androgen, estrogen, oxytocin and prolactin, adrenaline, corticosterone, and ACTH were purchased from Shanghai Jianglai Biological Technology Co., Ltd (Shanghai, China). The ELISA test was carried out according to the operation instructions of the kit, as follows:

(1) Dilute the antibody to 1–10 µg/mL with buffer solution. Add 0.1 mL to the reaction well overnight at 4 °C. The next day, discard the solution in the hole and wash it three times with washing buffer.

(2) Add sample: add 0.05 mL diluted sample to the coated reaction well and incubate at 37 °C for 1 h. Then wash (blank well, negative control well and positive control well at the same time).

(3) Add enzyme-labeled antibody: add 0.05 mL of freshly diluted enzyme-labeled antibody (titrated dilution) into each reaction well. Incubate at 37 °C for 0.5–1 h, then wash for three times.

(4) Add substrate solution for color: add 0.1mL TMB substrate solution temporarily prepared in each reaction well, 37 °C for 10–30 min.

(5) Stop the reaction: add 2M sulfuric acid 0.05 mL into each reaction well.

(6) Results: on a white background, directly observe the results with naked eyes: the darker the color of the reaction hole, the stronger the positive degree, the negative reaction is colorless or very light, according to the depth of the color, to "+", "-" sign. The OD value can also be measured: on the ELISA tester, at 450 nm, the blank control well is zeroed and then the OD value of each well is measured. If the OD value is greater than 2.1 times of the specified negative control, it is considered as positive.

## Data statistical analysis

Data statistical analysis were carried out using SPSS 25.0. Data is presented as mean $\pm$ SD. Homogeneity of variance test and normal distribution test were used to evaluate whether the data met the assumptions. Two-way ANOVA were used to analyze the changes of transmitter and hormone levels between two gender mice before and after chronic stress treatment. All analyses were made with a 95% CI, and the level of significance was set at $p < 0.05$. P values were $*p < 0.05$, $**p < 0.01$.

## RESULTS

In general, there were significant differences in behavior, neurotransmitters, sex hormones and stress hormones between male and female mice after 8 weeks of chronic stress model establishment ($p < 0.05$). In particular, the between-subjects effects analysis showed that there were significant interaction between gender and stress intervention on monoamine

**Table 2  Summary of test of between-subjects effects in monoamine transmitter and related hormone.**

| Items | Test of between-subjects effects ($p$ value) | | |
|---|---|---|---|
| | Gender | Stress | Gender*Stress |
| Exploratory behaviors scores | 0.511 | 0.0001 | 0.147 |
| Locomotion activity scores | 0.07 | 0.0001 | 0.012 |
| sucrose preference | 0.476 | 0.0001 | 0.074 |
| 5-HT | 0.002 | 0.0001 | 0.001 |
| Norepinephrine | 0.0001 | 0.03 | 0.011 |
| Dopamine | 0.047 | 0.0001 | 0.837 |
| Androgen | 0.123 | 0.017 | 0.332 |
| Estrogen | 0.043 | 0.0001 | 0.004 |
| Oxytocin | 0.0001 | 0.0001 | 0.043 |
| Prolactin | 0.0001 | 0.0001 | 0.004 |
| Adrenaline | 0.381 | 0.001 | 0.765 |
| Corticosterone | 0.734 | 0.0001 | 0.156 |
| ACTH | 0.636 | 0.0001 | 0.914 |

transmitter (5-HT, norepinephrine) and sex hormones (estrogen, oxytocin, prolactin) ($p < 0.05$) (Table 2).

## Verification of chronic stress-induced depression-like phenotype

Behavior assessments confirmed that the chronic stress-induced depression-like phenotype was established successfully. As shown in Figs. 1A & 1B, locomotion activity scores and exploratory behavior scores in stress mice were significantly decreased compared to control group after stress model establishment ($p < 0.05$). Further analysis demonstrated that sex difference was found: the locomotion activity scores in male stress mice declined more than in female stress mice (18.76% vs 7.82%, $n = 15$), while the exploratory behavior scores in female stress mice declined more than in male stress mice (35.86% vs 46.79%, $n = 15$). There was an interaction between gender and stress intervention on the effect of locomotion activity scores ($p = 0.012$). The sucrose preference percentage of two groups both declined , further analysis displayed that there was a sex difference of the decline range for male and female stress mice respectively (21.91% vs 14.24%, $n = 15$), as shown in Fig. 1C.

## Sex difference in monoamine transmitter after model establishment

After 8 weeks of chronic stress model establishment, monoamine transmitters such as 5-HT, dopamine, and norepinephrine in serum decreased in chronic stress group mice as compared to the baseline. There was an interaction between gender and stress intervention on the effect of 5-HT ($p = 0.001$) and norepinephrine ($p = 0.011$). As shown in Figs. 2A, 2B, the level of 5-HT in male stress mice decreased more than that in female stress mice (41.19% vs 27.23%, $n = 12$), while the level of norepinephrine in male stressed mice decreased less than that in female stress mice (38.83% vs 59.17%, $n = 12$). As for dopamine level, the reduction was slightly in male and female mice (9.51% vs 9.78%, $n = 12$) but there

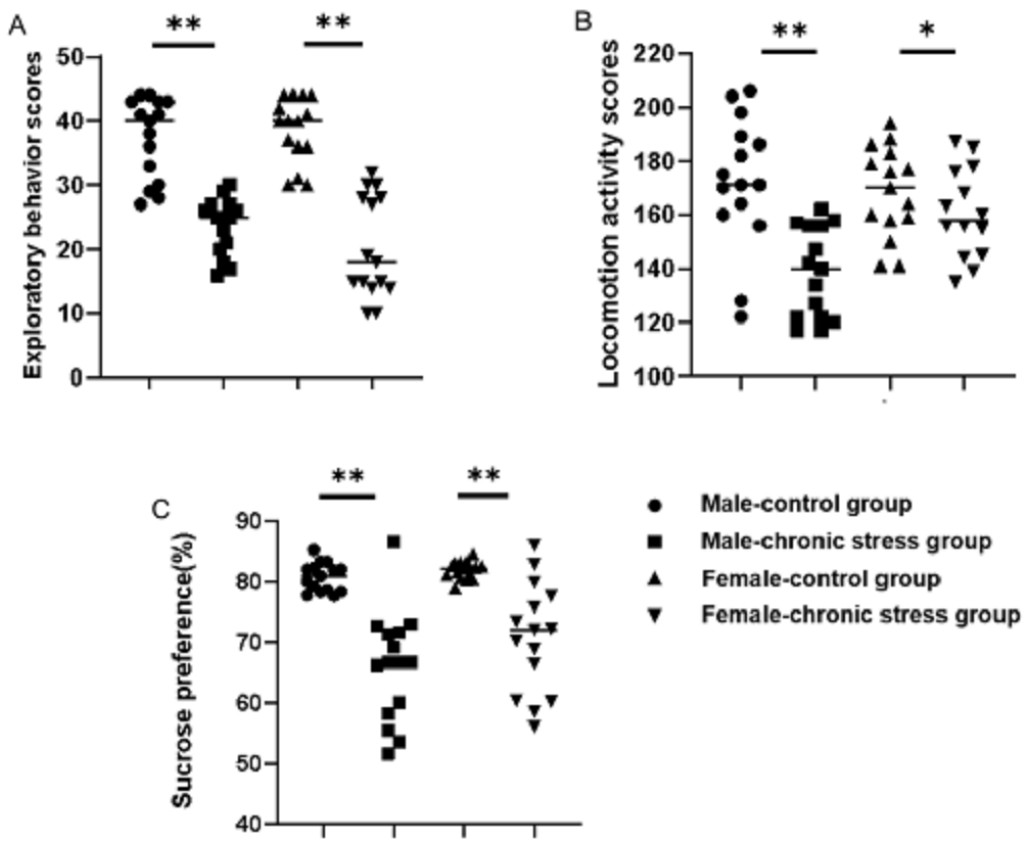

**Figure 1 Behavior assessment comparison after chronic stress model establishment in same sex.** Behavior changes assessment after chronic stress model establishment. (A) Exploratory behavior scores, $n = 15$; (B) locomotion activity scores, $n = 15$; (C) sucrose preference, $n = 15$. $*p < 0.05$, $**p < 0.001$.

was no statistical significance of dopamine between the male and female mice ($p = 0.837$), as shown in Fig. 2C.

## Sex difference in sex hormone after model establishment

After stress model establishment, sex hormones such as androgen, estrogen, oxytocin, and prolactin in serum were significantly decreased in chronic stress group mice compared with baseline. Interaction between gender and stress intervention existed in estrogen ($p = 0.004$), oxytocin ($p = 0.043$) and prolactin ($p = 0.004$). There was no an interaction between gender and stress intervention on the effect on androgen ($p = o.332$). Androgen (Fig. 3A), estrogen (Fig. 3C), and prolactin (Fig. 3D) levels in male stress mice decline less than those in female stress mice (13.42% *vs* 32.64%, 45.75% *vs* 69.32%, and 16.12% *vs* 23.14%, respectively, $n = 10$). Meanwhile the oxytocin (Fig. 3B) in male stress mice decreased more than in female stress mice (33.11% *vs* 17.11%, $n = 10$).

## Sex differences in stress hormone after stress model establishment

After stress model establishment, adrenaline, corticosterone, and ACTH were significantly increased in chronic stress group mice compared with baseline, whereas there was no

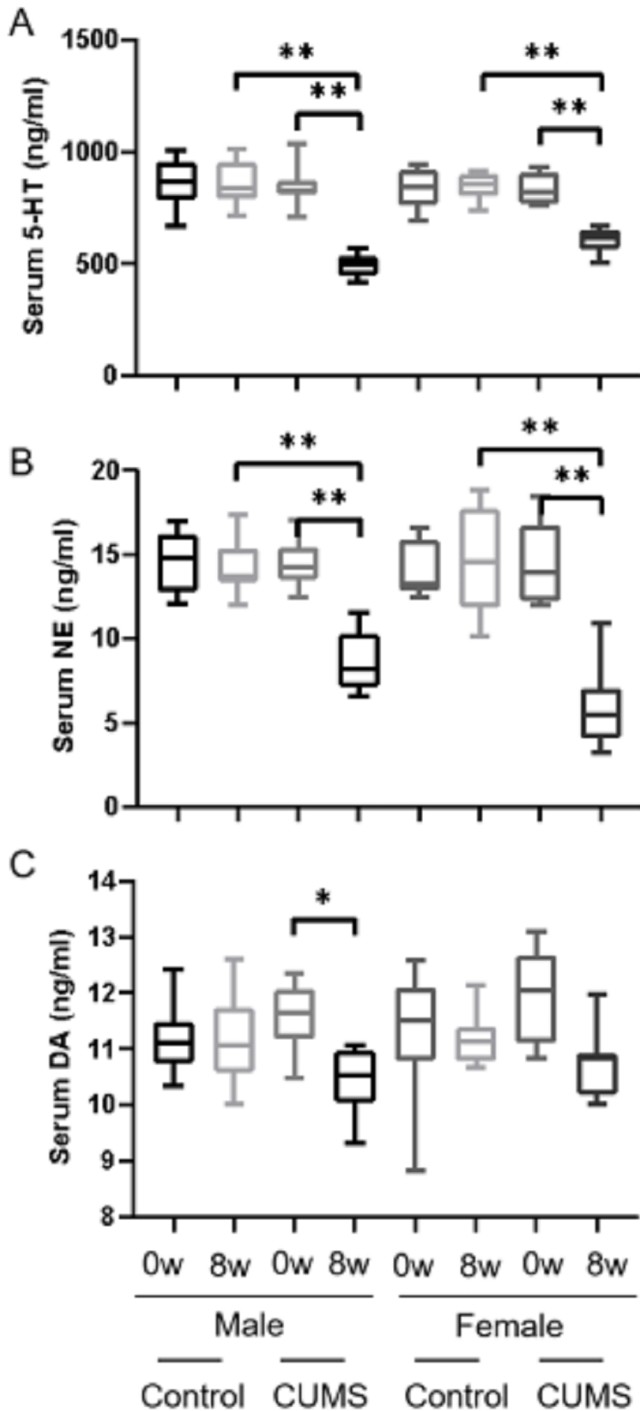

**Figure 2** **Monoamine transmitter levels before and after CUMS model establishment between two groups with different genders.** Monoamine transmitter levels before and after CUMS model establishment between two groups with different genders. (A) Serum 5-HT concentration, $n = 12$; (B) serum NE concentration, $n = 12$; (C) serum DA concentration, $n = 12$. *$p < 0.05$, **$p < 0.001$.

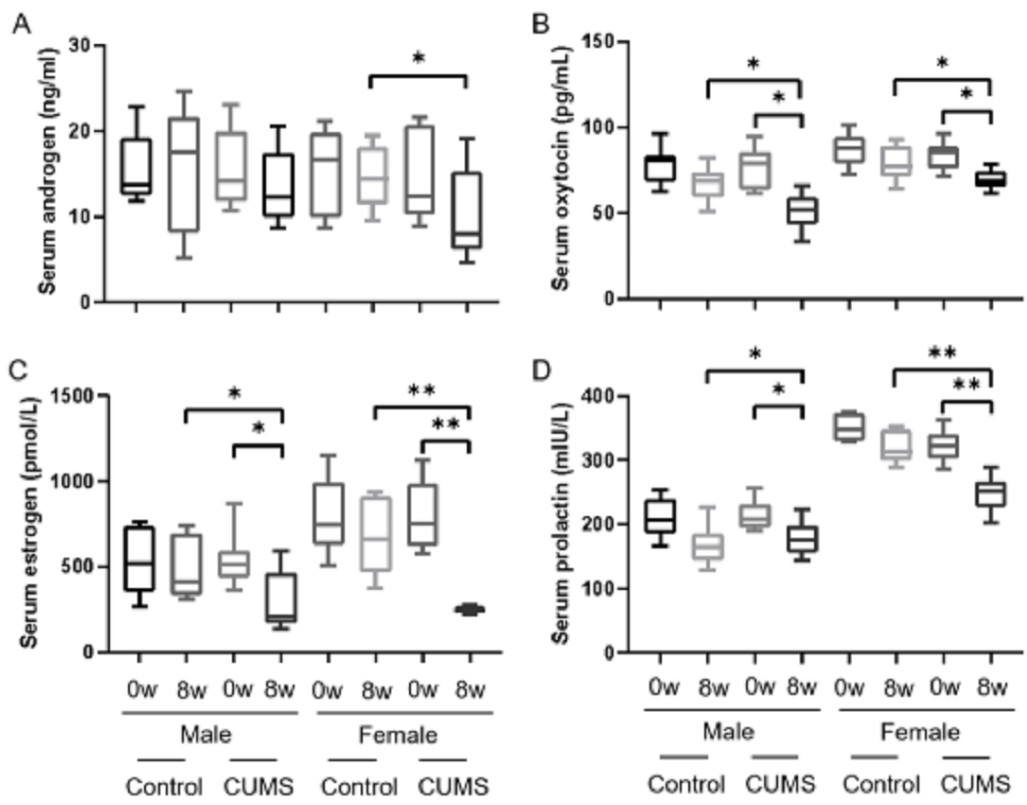

**Figure 3** **Sex hormone levels before and after CUMS model establishment in the control group and in chronic stressed group both in male and in female mice.** Sex hormone levels before and after CUMS model establishment in different group. (A) Serum androgen concentration, $n = 10$; (B) serum oxytocin concentration, $n = 10$; (C) serum estrogen concentration, $n = 10$; (D) serum prolactin concentration, $n = 10$. $^{*}p < 0.05$, $^{**}p < 0.001$.

interaction between gender and stress intervention. As shown in Fig. 4, serum adrenaline (Fig. 4A) in male mice increased more than in female mice ($p = 0.765$) (24.42% vs 22.22%, $n = 10$) and ACTH (Fig. 4C) in both male and female mice ($p = 0.914$) increased similarly (51.96% vs 51.81%, $n = 10$). However, the increasing amplitude of corticosterone (Fig. 4B) in male stress mice was more than that in female stress mice ($p = 0.156$) (38.46% vs 21.80%, $n = 12$).

## DISCUSSION

The strain of C57BL/6 mice was chosen for study due to their behavioral profile, such as moderate to high levels of social pattern, reversal learning and exploration behavior (*Moy et al., 2007*). In recent years, C57BL/6 mice have been widely used in behavioral studies (*An et al., 2011*; *Kubera et al., 1998*; *Yao et al., 2016*), and used as research objects for depression to explore the changes of biochemical molecule levels (*Kolbasi et al., 2021*; *Li & Singh, 2014*). A few research displayed some sex differences in behavior and hormone secretion but lack of comprehensive biochemical molecules profile elaboration (*Dalla et*

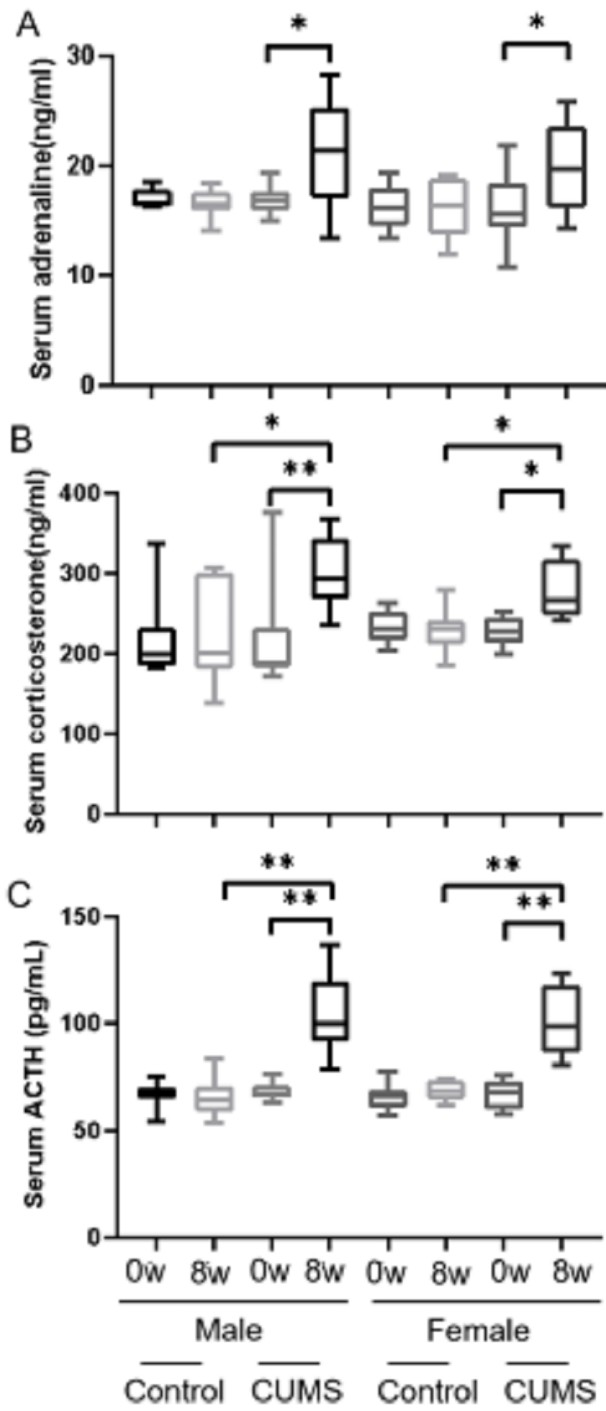

**Figure 4 Stress hormone levels before and after CUMS model establishment between two groups with different genders.** Stress hormone levels before and after CUMS model establishment between different gender. (A) serum adrenaline concentration, $n = 10$; (B) serum corticosterone concentration, $n = 12$; (C) serum ACTH concentration, $n = 10$. *$p < 0.05$, **$p < 0.001$.

*al., 2011*; *Goodwill et al., 2019*; *Huynh et al., 2011*; *Labaka et al., 2018*; *Leussis et al., 2012*; *Liu et al., 2019*).

In this study, we used C57BL/6 mice to establish a chronic stress-induced depression-like behavior mice model successfully at first, and then comprehensively detected the secretion of related transmitters, stress hormones and sex hormones in male and female mice. We found that sex differences of monoamine transmitter and related hormone levels in serum occurred in chronic stress induced depression-like phenotype mice model. Specifically, in transmitters and hormones, male mice showed more decline range in 5-HT and oxytocin, while female mice displayed more decline in norepinephrine, androgen, estrogen and prolactin. The increasing amplitude of corticosterone in male stressed mice was more than that in female stressed mice. As summarized in Table 3, although the same trend of the hormone as mentioned is revealed in male/female stress mice, sex difference still exists.

The monoamine hypothesis plays an important role in the pathogenesis of depression, which believes that depression is closely related to the low level of monoamine such as NE and 5-HT in brain. Although the serotonin hypothesis of depression is still influential, a recent systematic umbrella review of the principal relevant areas of research on whether depression is associated with lowered serotonin concentration or activity suggested that peripheral serotonin levels have no correlation with depression and might not be useful as a biomarker (*Frank et al., 2000*, *Moncrieff et al., 2022*). The clinical relevance of peripheral monoamine measure warrants further investigation.

As to sex hormones, our study revealed that chronic stress decreased androgen level both in male and female stress mice. The reduction of androgen in male mice was less than female mice (13.42% *vs* 32.64%), which suggests that androgen might play a role in depression development. Glucocorticoids signaling from the HPA axis can alter levels of these neurosteroids, demonstrating a reciprocal interaction between the two systems (*Finn, 2020*). The levels of androgen-derived neurosteroids were associated with increased negative affect. Previous studies indicated that there was positive correlation between androgen and depression symptom severity (*Hung et al., 2019*). Androgen deficiency was associated with greater depression and anxiety symptom severity (*Kimball et al., 2019*). In addition, we also found that estrogen levels decreased about 70% in female stress mice after chronic stress model establishment, which suggests that estrogen is engaged in depression development as like androgen. Loss of estrogen contributed to depression in perimenopause and beyond menopause (*Chen & Chen, 2021*). Premature ovarian insufficiency and related estrogen deficiency was a cause of many psychological symptoms including depression, psychological tension, and other mood disorder (*Słopień, 2018*). Monoaminergic loss caused by estrogen depletion was deemed to a potential mechanism explaining postpartum depressed mood (*Sacher et al., 2010*). Estradiol could be used to treat decreased serotonin response in postmenopausal women (*Halbreich et al., 1995*). Estrogen–serotonin interactions perhaps impact depression symptoms through the cognitive and mood regulatory functions of the serotonergic system. A potential role for estrogen in depression need us to pay more attention, including assessing the predictive benefit of estrogen effects on stress response and emotional cognition. In this study, there was an interaction between gender and

**Table 3  Summary of monoamine transmitter and related hormone change.**

| Class | | Change trend | Change extent (%) | |
|---|---|---|---|---|
| | | | Male | Female |
| Transmitter | 5-HT | ↓ | 41.19[*] | 27.23[*] |
| | Dopamine | ↓ | 9.51 | 9.78 |
| | Norepinephrine | ↓ | 38.83[*] | 59.17[*] |
| Sex hormone | Androgen | ↓ | 13.42[*] | 32.64[*] |
| | Estrogen | ↓ | 45.75[*] | 69.32[*] |
| | Oxytocin | ↓ | 33.11[*] | 17.11[*] |
| | Prolactin | ↓ | 16.12[*] | 23.14[*] |
| Stress hormone | Adrenaline | ↑ | 24.42[*] | 22.22[*] |
| | Corticosterone | ↑ | 38.46[*] | 21.80[*] |
| | ACTH | ↑ | 51.96[*] | 51.81[*] |

Notes.

[*]$P < 0.05$ compared to the corresponding score of the pre-model establishment.

stress on the secretion of estrogen, suggesting that estrogen may be used as an indicator of depression in different genders.

In this study, we also noted that oxytocin as well as prolactin levels decreased after 8 weeks of chronic stress both in female mice and more in male mice, which suggests that oxytocin monitor might also be meaningful for male patients with depression. Oxytocin might relieve stress and regulates mood. Early life stress can cause disorder of oxytocin expression, which may increase the risk of developing depression and other mental disorders in adulthood (*Luo et al., 2017*). The release of oxytocin might minimize the risk for depressive symptoms (*Badr & Zauszniewski, 2017*; *McQuaid et al., 2014*). Oxytocin has been shown to protect hippocampal neurons from the toxic effects of glucocorticoids (*Matsushita et al., 2019*). Changes in prolactin levels in affective disorders suggested the possibility of the prolactin as biochemical and neuroendocrinological biomarker in depression diagnosis. Still, there were a few contradictory data caused by different research paradigms measurement of peripheral prolactin serum levels–different basal conditions, different circadian pattern, or different challenges (*Ben Hadj Ali, 1987*; *Nicholas, Dawkins & Golden, 1998*).

Stress hormones were significantly increased in male and female mice after chronic stress in this work. Adrenaline and ACTH level were both increased in male and female mice to the similar extent. It is worth noting that the increasing amplitude of corticosterone in male stress mice was more than that in female stress mice. HPA axis was usually over-activated and the secretion of corticotropin releasing hormone (CRH) from neuroendocrine neurons of the PVN is increased as mice were exposed to stress (*Dunn & Berridge, 1990*). Altered HPA function may be important in the etiology of depression in women. The cortisol response to stress demonstrated sex differences (*Young & Korszun, 2010*) which affected by the menstrual cycle (*Kirschbaum et al., 1999*; *Kudielka & Kirschbaum, 2005*) and pregnancy (*Altemus, 2006*).

## CONCLUSIONS

Collectively, sex differences exist in the chronic stress mice model in respect of behavior, monoamine transmitter, sexual hormone, and stress hormone serum levels. The sex difference of monoamine transmitter and related hormone levels in peripheral serum might be reference for depression diagnosis of different sex in future.

### Sample size

According to literature and experimental results, behavioral scores of subjects were used as indicators to determine sample size in hypothesis testing. The mean behavioral scores of the control group were $173.14 \pm 21.84$, and the behavioral scores of the expected stress group decreased by 25 points, establishing a bilateral alpha is 0.05 and a confidence interval of 90%. According to the sample size calculation formula, the control group and the experimental group each need 15 mice.

### Study limitations

In this study, we only used a single species of C57BL/6 mice model to conduct the research. Due to the differences between different species, our results may not be universal. The changes of the transmitters and hormones displayed in mice model may not exactly extrapolated in human beings.

### Statement

The protocol was prepared before the study and was not registered.

### Funding

This work was supported by the Shanghai Municipal Science Commission (Grant 19411971700 to Feng Xu). The funders had no role in study design, data collection and analysis, decision to publish, or preparation of the manuscript.

### Grant Disclosures

The following grant information was disclosed by the authors:
Shanghai Municipal Science Commission: 19411971700.

### Competing Interests

The authors declare there are no competing interests.

### Author Contributions

- Yitian Chen conceived and designed the experiments, performed the experiments, prepared figures and/or tables, authored or reviewed drafts of the article, and approved the final draft.
- Weijia Cai conceived and designed the experiments, performed the experiments, authored or reviewed drafts of the article, and approved the final draft.
- Canye Li performed the experiments, authored or reviewed drafts of the article, and approved the final draft.
- Zuanjun Su performed the experiments, authored or reviewed drafts of the article, and approved the final draft.
- Zhijun Guo analyzed the data, authored or reviewed drafts of the article, and approved the final draft.
- Zhuman Li analyzed the data, authored or reviewed drafts of the article, and approved the final draft.
- Chen Wang performed the experiments, prepared figures and/or tables, authored or reviewed drafts of the article, and approved the final draft.
- Feng Xu conceived and designed the experiments, prepared figures and/or tables, and approved the final draft.

## Data Availability

The raw data is available in the Supplementary Files.

## Supplemental Information

Supplemental information for this article can be found online at http://dx.doi.org/10.7717/peerj.14014#supplemental-information.

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
