# Peer review of "Sex differences in peripheral monoamine transmitter and related hormone levels in chronic stress mice with a depression-like phenotype"

_PeerJ, doi:10.7717/peerj.14014_

## Round 0.1 · original submission · Major Revisions

I am sorry to inform you that your manuscript is not acceptable for publication as submitted. Substantial English editing and a number of additional experiments are deemed necessary to substantiate your hypothesis.

·

Basic reporting

Literature references are topical and well used. However context could be improved, some introduction on the previous study's methods would help the reader understand the limits of those previous studies. Also more specific notation of the scope of depression's effects on the populace.

Figures will need to be clarified with data presentation, graphs do not note n-values or show individual scores that will provide clarity in presentation.

I would suggest more detail be provided on the specific relationship between these specific elements and testosterone or estrogen. Such as are any of them downstream of relevant promoters or is the fuel for creating these hormones altered by exposure to sex hormones.

Experimental design

No comment.

Validity of the findings

Additional testing is required to determine if the mice suffer from depression specifically. The standard criteria includes learned helplessness, which the experimental model does not demonstrate. Anxiety which can also accompany hesitancy in exploration and a lack of pleasure seeking behaviors. If possible treatment with an SSRI may reinforce the hypothesis of depression.

Reviewer 2 ·

Basic reporting

In the manuscript submitted by the authors for review, the sex differences of C57BL/6 mice were described. Each sex was divided into control groups and subjected to CUMS. Then, the behavioral changes and some biochemical parameters were assessed in the serum of these animals.

Unfortunately, the manuscript is difficult to understand linguistically. The English correction of the text is necessary.

Experimental design

The brief introduction presents information about depression and its connection with stress. However, no reference was made to the literature data on the measured biochemical parameters, and a lot is known about it. First of all, research is carried out on animals, not on humans, and such works should be sought for introduction. It is also unknown why this particular mouse strain was chosen. A few words about the stress response of mice of this strain should be included in the introduction. However, the authors did not refer to the literature concerning the measured biochemical parameters, and a lot is known about this. First of all, research is performed on animals, not humans, and such literature should be searched and included in the introduction. It is also unknown why this particular mouse strain was chosen. A few words about the stress response of mice of this strain should be included in the introduction.

Validity of the findings

The materials and methods are described quite comprehensively for the behavioral part. However, the biochemical part has been described very generally. The description of the ELISA carrying out should be more detailed. I also have doubts about blood samples clotting for 2 hours, it should cause high hemolysis. Please check what parameters are provided in the Elise manual in this regard.
The results were described without reference to specific graphs. It was not included in the text the parts ( A, B, C…) of graphs. The results are presented as the difference in the stress response of males compared to females. In order to compare in this way the experimental data from a given measured parameter should be included in the same graph for males and females. Then it is easy for the reader to notice the described changes.
The charts are too small, so it's hard to see which chart applies to which change. Please increase the charts or descriptions on them.

Additional comments

It is not clear, from the discussion, why these tests were performed. For example, are the results new or do they duplicate other changes already shown? Is there any literature (connected to animals) to which the results can be referenced? Are there any results obtained on other strains of mice and those presented by the authors are new and pioneering? What is the main achievement of the research? What the authors wanted to achieve by performing such research? If the authors wanted to determine new biomarkers of depression, they should first of all refer to the manuscripts of depression or stress biomarkers. I believe, that the discussion needs to be rewritten.

---

## Round 0.2 · Major Revisions

Unfortunately your revisions were not consider adequate to warrant publication. Given the nature of the comments, I will consider one more revision. If however, you decide to submit elsewhere please let me know.

Reviewer 2 ·

Basic reporting

Thank you for your reply to the review. Some of my comments were taken into account.
However, I have a few more comments:
• The manuscript still requires linguistic correction. Sentences: ‘The decrease of norepinephrine, androgen and estrogen in female stressed mice showed more than in male stressed mice.’ should be written: ‘The norepinephrine, androgen and estrogen decreased more in female stressed mice than in male stressed mice.’ e.t.c. Additionally, there are linguistic errors in the text.

Experimental design

• There are appropriate statistical tests, that should be used to show significant differences between the groups. In this case, a two-way ANOVA, where one factor is gender and the other is the presence or absence of stress, or a three-way ANOVA if the time factor is additionally taken into account. Such tests have not been carried out.

Validity of the findings

• I propose to rethink the construction of the figures. Are you sure, the baseline level is needed in the charts?

Additional comments

• While the discussion was expanded, it is still unknown why such strain of mice was selected for the study, and what this research was intended to bring to the science. The authors write: 'Previous data displayed some sex differences in behavior but lack of biomarker information.' However, I propose a revision of the literature in order to find papers about biochemical analyzes in murine models of stress and depression. They are really available to the strain of mice selected by the authors.

---

## Round 0.3 · accepted · Accept

Thank you for making significant changes to your manuscript as it is now deemed acceptable for publication. Thank you for choosing PeerJ and we hope that you will consider the journal for future manuscripts.

·

Basic reporting

no comment

Experimental design

no comment

Validity of the findings

no comment

Additional comments

The citation of work by Moncrieff is currently noted by many as being controversial for being frequently overstated and politicized. Her interpretations have come under much scrutiny in the community as a consequence. The work by Jones depicts the points you are making quite clearly. I would not necessitate removal of her work as a reference but might note the status of the work to ensure to the audience that the author is aware.